# Bivalirudin vs. Enoxaparin in Intubated COVID-19 Patients: A Pilot Multicenter Randomized Controlled Trial

**DOI:** 10.3390/jcm11205992

**Published:** 2022-10-11

**Authors:** Eugenio Garofalo, Gianmaria Cammarota, Giuseppe Neri, Sebastiano Macheda, Eugenio Biamonte, Pino Pasqua, Maria Laura Guzzo, Federico Longhini, Andrea Bruni

**Affiliations:** 1Anesthesia and Intensive Care Unit, Department of Medical and Surgical Sciences, “Magna Graecia” University, 88100 Catanzaro, Italy; 2Department of Anesthesia and Intensive Care Medicine, University of Perugia, 06121 Perugia, Italy; 3Anesthesia and Intensive Care Unit, Grande Ospedale Metropolitano, 89121 Reggio Calabria, Italy; 4Anesthesia and Intensive Care Unit, Annunziata Hospital, 87100 Cosenza, Italy; 5Anesthesia and Intensive Care Unit, “Pugliese Ciaccio” Hospital, 88100 Catanzaro, Italy

**Keywords:** COVID-19, bivalirudin, enoxaparin, anticoagulation, critically ill patient, acute respiratory failure

## Abstract

(1) Background: In COVID-19 patients, the occurrence of thromboembolic complications contributes to disease progression and mortality. In patients at increased risk for thrombotic complications, therapeutic enoxaparin should be considered. However, critically ill COVID-19 patients could develop resistance to enoxaparin. Bivalirudin, a thrombin inhibitor, may be an alternative. This pilot multicenter randomized controlled trial aims to ascertain if bivalirudin may reduce the time spent under invasive mechanical ventilation, as compared to enoxaparin. (2) Methods: Intubated COVID-19 patients at risk for thrombo-embolic complications were randomized to receive therapeutic doses of enoxaparin or bivalirudin. We ascertained the time spent under invasive mechanical ventilation during the first 28 days from Intensive Care Unit (ICU) admission. A standardized weaning protocol was implemented in all centers. In addition, we assessed the occurrence of thromboembolic complications, the number of patients requiring percutaneous tracheostomy, the gas exchange, the reintubation rate, the ICU length of stay, the ICU and 28-days mortalities. (3) Results: We enrolled 58 consecutive patients. Bivalirudin did not reduce the time spent under invasive mechanical ventilation as compared to enoxaparin (12 [8; 13] vs. 13 [10; 15] days, respectively; *p* = 0.078). Thrombotic (*p* = 0.056) and embolic (*p* = 0.423) complications, need for tracheostomy (*p* = 0.423) or reintubation (*p* = 0.999), the ICU length of stay (*p* = 0.076) and mortality (*p* = 0.777) were also similar between treatments. Patients randomized to bivalirudin showed a higher oxygenation at day 7 and 15 after randomization, when compared to enoxaparin group. (4) Conclusions: In intubated COVID-19 patients at increased risk for thromboembolic complications, bivalirudin did not reduce the time spent under invasive mechanical ventilation, nor improved any other clinical outcomes.

## 1. Background

Since December 2019, a novel coronavirus (SARS-CoV-2) has spread worldwide, causing a new disease (COVID-19) characterized by a dysregulated response of the immune system and Acute Respiratory Failure [1]. The exaggerated inflammatory activity generates a prothrombotic condition [2], promoting vascular thrombosis, microangiopathy and occlusion of small vessels in the lungs [3] and in other organs [4,5,6]. The development of thrombo-embolic complications contributes to the disease progression and lethality [2,7]. For these reasons, immunomodulant and/or anticoagulant drugs have been proposed in the treatment of COVID-19 since the beginning of the pandemic [8,9,10,11].

The recent ESCMID guidelines suggest the administration of low molecular weight heparin at prophylactic doses for the prevention of venous thromboembolism [12]. In critically ill COVID-19 patients, a large trial suggests that an initial strategy of therapeutic anticoagulation with heparin does not improve hospital survival and length of organ support as compared to prophylactic doses [13]. However, therapeutic anticoagulation with heparin should be considered only in patients at high risk for thromboembolic complications, on an individual basis [14]. The administration of enoxaparin is associated with lower intubation rate, hospital mortality [15], and improved 28 day-survival [16]. Nonetheless, in the literature it has been reported that up to 50% patients has a high factor Xa residual activity and may be resistant to enoxaparin [17,18], leading to thrombocytopenia, arterial and venous thrombosis [19].

Bivalirudin is a short (25 min) half-life time direct inhibitor of both free-circulating and fibrin-bound thrombin. As opposed to low molecular weight heparin, bivalirudin does not induce resistance or thrombocytopenia [20]. For these advantages, bivalirudin has been already successfully used in COVID-19 patients requiring Extra-Corporeal Membrane Oxygenation (ECMO) [20,21].

In this scenario, bivalirudin may be a valid alternative to enoxaparin to guarantee the therapeutic anticoagulation in critically ill COVID-19 patients at high risk for thromboembolic complications and to prevent thromboembolic complications. We have therefore designed this pilot, off-label randomized controlled trial to assess if the infusion of bivalirudin may reduce the time spent under invasive mechanical ventilation, as compared to enoxaparin (first outcome). We have also ascertained whether bivalirudin may diminish the incidence of thrombotic and/or thromboembolic complications, improve gas-exchange and reduce the ICU length of stay and mortality (additional outcomes).

## 2. Materials and Methods

This pilot, open label randomized controlled trial was conducted in four (60 beds) tertiary referral center Intensive Care Units (ICU) of the “Mater Domini” University Hospital and “Pugliese Ciaccio” Hospital in Catanzaro (Italy), “Annunziata” Hospital in Cosenza (Italy) and “Grande Ospedale Metropolitano” in Reggio Calabria (Italy), after local Ethical committee approval (Ethical Committee Approval number 476 on 17th December 2020). Written informed consent was obtained from patients or their next of kin, also for publication of their individual details and accompanying images in the manuscript. This trial was compliant with the Consolidated Standards of Reporting Trials (CONSORT) reporting guidelines and was prospectively registered (NCT05334654; www.clinicaltrials.gov, accessed on 13 April 2022).

### 2.1. Population

We screened for eligibility all consecutive adult (i.e., >18 years/old) patients admitted in ICU and requiring invasive mechanical ventilation (since the previous 24 h) for SARS-CoV-2 pneumonia. SARS-CoV-2 infection was ascertained through polymerases chain reaction nasal swab. Inclusion criteria were: (1) acute onset (within 1 week) of respiratory failure; (2) arterial tension (PaO_2_) to inspiratory oxygen fraction (FiO_2_) ratio (PaO_2_/FiO_2_) < 200 mmHg; (3) evidence of bilateral pulmonary infiltrates at chest imaging (i.e., X-ray or computed tomography scan); (4) plasma D-dimer level greater than 4 times the upper limit of normal based on local laboratory criteria (normal range 0 to 0.55 mg/L) [22]. Of note, the last criterion defines a population at high risk of thromboembolic complications and worsened outcome, requiring therapeutic anticoagulation [22].

Patients were excluded if meeting one or more of the following exclusion criteria: (1) history of bleeding within the previous month; (2) cerebral, thoracic or abdominal surgery in the previous 15 days; (3) active gastrointestinal or intracranial cancer; (4) hepatic dysfunction with baseline International Normalized Ratio (PT-INR) >1.5 [22]; (5) creatinine clearance <30 mL/min/1.73 m^2^; (6) platelet count <25,000/μL [22]; (7) history of heparin-induced thrombocytopenia (HIT) within 100 days [22]; (8) hypersensitivity/intolerance to study drug or components; (9) need for urgent ECMO run; (10) presence of thrombotic or thromboembolic complications and 11) denied consent.

### 2.2. Randomization

At enrollment, patients were randomized into either the intervention (bivalirudin) or control (enoxaparin) group. Randomization was achieved through a computer-generated random sequence, obtained by an independent investigator, not otherwise involved in the trial, with an allocation ratio of 1:1 and with a permuted block method. A single randomization list was created for all participants centers. Allocation concealment was obtained using sequentially numbered sealed opaque envelopes. Each envelope contained the allocation of the patient to either intervention (bivalirudin) or control (enoxaparin) group, with a unique patient identifier code. The randomization was based on a centralized phone call system. Due to the research design, neither the individual collecting data nor the patient could be blinded to treatment allocation.

### 2.3. Treatments

All patients received standard care, according to the current clinical practice guidelines and evidence-based recommendations/indications [23,24].

Patients randomized in the control group received subcutaneous enoxaparin 100 UI/Kg twice daily if Glomerular Filtrate Rate (GFR) was between 60–120 mL/min/1.73 m^2^, whereas it was reduced to 50 UI/Kg twice daily if the GFR was 60–30 mL/min/1.73 m^2^.

Patients randomized in the intervention group received bivalirudin in continuous infusion until ICU discharge. After a bolus of 0.25 mg/kg of bivalirudin in 30 min, continuous infusion was set at 0.2 mg/kg/h. The infusion rate was then adjusted targeted to an activated Partial Thromboplastin Time (aPTT) of about 60 s; coagulation was ascertained every 8 h through a dedicated system (BCS^®^ XP System, Siemens Healthcare Diagnostic, Inc., Milano, Italy).

### 2.4. Discontinuation of Study Drugs

Discontinuation of study treatments was considered when one of the following conditions occurred: (1) withdrawal of the consent to participate to the study; (2) occurrence of life-threatening adverse events suspected to be related to the trial medication and/or prevents patient’s continuation on study medication (i.e., uncontrolled bleeding); (3) an investigator considered it advisable for explicit and documented clinical reasons.

### 2.5. Study Endpoints

The primary study endpoint was the time spent under invasive mechanical ventilation during the first 28 days of ICU admission. Noteworthy, the weaning protocol was conducted with invasive mechanical ventilation, as previously described [25,26,27]. In particular, once able to trigger the ventilator, patients were switched from volume controlled to pressure support ventilation. Throughout the weaning process, the FiO_2_ and positive end-expiratory pressure were set to maintain peripheral oxygen saturation (SpO_2_) between 92 and 96%, while the inspiratory pressure support was titrated to generate a tidal volume of 6–8 mL/kg of ideal body weight. Once ready, the spontaneous breathing trial was conducted with a low (2 cm/H_2_O) positive end-expiratory pressure with no inspiratory support for 30 min. Criteria for spontaneous breathing trial success or failure are those previously reported [25,26,27]. Patients who passed the spontaneous breathing trial were immediately extubated and non-invasive ventilation applied through a mask or helmet for the next 12 h, to avoid the occurrence of post-extubation respiratory failure [28]. In case of post-extubation respiratory failure, patients were reintubated. High-Flow Nasal Cannula were also suggested over standard oxygen therapy whenever required [29,30]. In addition (secondary outcomes), we assessed the occurrence of thrombotic or thromboembolic complications, the number of patients requiring percutaneous tracheostomy, the gas exchange as assessed through arterial blood gases (ABGs), the 28-free ventilation days [31], the reintubation rate, the ICU length of stay, the ICU and 28-days mortality.

### 2.6. Data Collection and Outcome Assessment

For each patient, we collected anthropometric and the following clinical baseline characteristics: presence of comorbidities, Simplified Acute Physiology Score (SAPS) II, Sequential Organ Failure Assessment (SOFA) score, Padua prediction score for venous thromboembolism [32,33,34], vital parameters (heart rate, arterial pressure), arterial blood gases (ABGs). Furthermore, we recorded the following blood test: platelet count, aPTT, PT-INR, D-Dimer and Fibrinogen, C-Reactive Protein and Procalcitonin.

ABGs and blood tests were recorded also at Day 3, 7 and 15 after the ICU admission.

Daily, until ICU discharge, all patients underwent a color-coded Doppler ultrasonography to assess the presence of thrombi in both superficial and deep veins of the upper and lower limbs, and in the internal jugular veins. In case of suspicion, Computed Tomography pulmonary angiogram was performed to exclude or confirm the presence of pulmonary embolism.

### 2.7. Statistical Analysis

The primary outcome was to assess if bivalirudin might reduce the time spent under invasive mechanical ventilation within 28 days, as compared to enoxaparin. A previous Italian study reported that the median [interquartile range] duration of invasive mechanical ventilation in COVID-19 patients was 10 [6–17] days [35]. The estimated mean of time spent under invasive mechanical ventilation was 11 (8) days [36]. We arbitrarily assumed that bivalirudin may reduce the time spent under invasive mechanical ventilation by 3 days, with an expected standard deviation of 5 days. Considering an alpha error of 5% and a power of 80%, a total of 45 patients were needed to assess our study aim. Assuming a 10% of patients drop out and an increase in sample size for non-parametric analysis of 15%, the final sample size consisted of 58 patients, 29 per group respectively.

The normal data distribution was assessed via the Kolmogorov-Smirnov test. An intention-to-treat analysis was used. Results have been expressed as mean (standard deviation) or median (interquartile range 25–75), on the basis of data distribution. Continuous data have been compared with the Mann-Whitney U-test or the *t*-test of Student for unpaired data. The categorical variables were evaluated with the McNemar or Fisher exact test. We also determined the Kaplan-Meier curves, depicting the two groups for: (1) the time from intubation to liberation from invasive mechanical ventilation, (2) the ICU and (3) the 28-day survival; curves were also compared using the log-rank test. We considered significant two-sided *p* values < 0.05. Statistical analysis was performed using the Sigmaplot v. 12.0 (Systat Software Inc., San Jose, California, CA, USA).

## 3. Results

We enrolled 58 consecutive adult patients from April to June 2022. The enrollment flowchart is reported in Figure 1. All patients received the allocated treatment without protocol deviation or interruption. Characteristics of the patients at ICU admission and randomization are listed in Table 1. The two study groups were balanced according to demographic, anthropometric and clinical features of the patients at randomization. All patients received corticosteroids during the ICU length of stay, whereas none received antiviral therapies or tocilizumab, being all intubated, according the indications of the Italian drug agency.

### 3.1. Primary Outcome

Data on the primary outcome are available for all patients. The time spent under invasive mechanical ventilation during the first 28 days of ICU admission were 12 [8,13] days in the bivalirudin group and 13 [10,15] days in controls (i.e., enoxaparin group) (*p* = 0.078). The Kaplan-Meier curve indicating the time from intubation to liberation from invasive ventilation is depicted in Figure 2. The comparison of survival showed a log-rank *p* = 0.031.

### 3.2. Secondary Outcomes

Table 2 shows the secondary outcomes. Thromboembolic complications were higher in the enoxaparin groups (*p* = 0.012). On the opposite, the number of thrombotic (*p* = 0.056) and embolic (*p* = 0.423) complications were similar between patients’ groups, if analyzed separately. Need to tracheostomy (*p* = 0.423) or reintubation (*p* = 0.999) were also comparable between treatment groups. The 28-free ventilation days, ICU length of stay and mortality, and the 28-day mortality rate did not also differ between patients receiving bivalirudin or enoxaparin. Noteworthy, no patients showed bleeding complications in both groups.

Kaplan-Meier curve indicating the 28-free ventilator days, ICU and 28-day survival are depicted in Figure 3, Figure 4 and Figure 5, respectively. The comparison of survival curves did not show any difference between groups with respect to both outcomes (log-rank *p* = 0.085 for 28-free ventilator days, *p* = 0.060 for ICU survival curves and *p* = 0.640 for the 28-day survival curves).

Hemodynamic and ventilatory supports, ABGs and blood tests from Day 3, 7 and 15 are shown in Appendix A, respectively. Of note, at day 7 and 15 after randomization, PaO_2_/FiO_2_ was higher in patients receiving bivalirudin, as compared to those randomized to Enoxaparin. All the remaining results are similar between study groups. Lastly, 8 (28%) patients in the bivalirudin group and 14 (48%) patients in the control group developed a septic shock (*p* = 0.176).

## 4. Discussion

This pilot randomized controlled trial fails to demonstrate that infusion of bivalirudin significantly reduces the time spent under invasive mechanical ventilation, as compared to enoxaparin at therapeutic doses in critically ill COVID-19 patients at high risk of thromboembolic complications. In addition, bivalirudin did not improve any additional clinical outcome, including the incidence of thrombotic and/or thromboembolic complications, gas-exchange, need for tracheostomy, reintubation rate, the ICU length of stay and the ICU and 28-day mortalities.

Although a small pilot study, this is the first trial randomizing COVID-19 patients undergoing invasive mechanical ventilation and considered at risk for thromboembolic complications to receive enoxaparin (standard therapy) or bivalirudin (intervention group). The evidence in favor of bivalirudin in critically ill patients is growing, especially in patients requiring extracorporeal support to reduce the risk for thrombosis [37,38,39].

An observational study by Pieri et al. enrolled 129 patients, 46 receiving a prophylactic dose of heparin, 60 therapeutic heparin and 23 a full dose of bivalirudin [40]. When compared to therapeutic doses of heparin, the bivalirudin group was characterized by a longer ICU length of stay and higher number of major bleeding episodes and need for red blood cell transfusions. No difference was recorded with respect to ICU mortality. Noteworthy, patients receiving bivalirudin were more severe, as demonstrated by a higher SAPS-II score (60 vs. 39) and a higher need for ECMO (65%) [40]. Beyond the higher quality (randomized vs. observational) design, our two groups are homogeneous providing stronger evidence. Our cohorts of patients are similar for severity (i.e., SAPS-II score) to the bivalirudin group of the study by Pieri et al. [40]. However, some clinical outcomes are different between studies. The mortality reported by Pieri et al. in the bivalirudin group was 57% [40], whereas we report 27.6%. This large difference may be explained by the higher need for ECMO and major bleeding complications and transfusions, which we have not recorded.

Our study aimed to demonstrate a 3-day reduction of time spent under invasive mechanical ventilation in the bivalirudin group, as compared to patients receiving high-doses of enoxaparin. By expecting a reduction of pulmonary thromboembolic complications, we hypothesized that the time spent under mechanical ventilation would be significantly reduced. Recently, McCall et al. reported that severe right ventricular dysfunction occurs in up to 6% of mechanically ventilated COVID-19 patients and it is associated with an increased mortality [41]. Furthermore, the oxygenation is largely impaired and ventilation is more challenging (higher plateau airway pressure, lower dynamic compliance, higher need for prone positioning) [41]. Although a non-significant trend has been recorded and the Kaplan-Meier curve indicates that the probability to be under invasive mechanical ventilation is reduced in the bivalirudin group, the trial must be considered negative for the first outcome. This result may be explained by several reasons. First of all, the sample size is underpowered to demonstrate a statistically significant difference in the first study outcome and the hypothesized difference is too large. However, this trial must be considered as pilot study and further trials should be conducted. Second, COVID-19 patients may face complications other than pulmonary thromboembolism which may prolong the time spent under invasive mechanical ventilation. We have recently reported that the incidence of ESKAPE multidrug resistant bacteria is higher in critically ill COVID-19 patients as compared to non-COVID-19 patients [42]. The occurrence of sepsis and septic shock by multidrug resistant bacteria significantly prolongs the time spent under mechanical ventilation and the ICU length of stay [42]. Therefore, the effect of bivalirudin could have been partially hidden by this factor. This holds true also if we highlight that the study by Grasselli et al. reports a median time spent under invasive mechanical ventilation of 10 [6–17] days [42], whereas we recorded longer periods.

Before drawing our conclusions, our study also has limitations that have to be addressed.

First, the sample size population is probably underpowered for our study aim, as we have already discussed above. Another pilot study aimed to assess a reduction of the time spent under invasive mechanical ventilation enrolled even a lower number of patients [43]; then, the following randomized controlled trial recalibrated the sample size to re-assess the study aim [27].

Second, this is a pilot study that cannot be used to provide any clear evidence in favor or against the use of bivalirudin in this population. Therefore, a dedicated larger randomized controlled trial should be designed for this purpose.

Third, the trial has a multicenter design. This is a strength if we considered the applicability of the treatment in more centers and the reduction of the selection bias. However, the first study aim would have been influenced by different ventilatory and weaning strategies among centers. Noteworthy, the ventilatory strategy and weaning protocol were predefined based on current literature [25,26,27] and strictly followed by all centers.

Fourth, anticoagulation with enoxaparin was administer with a fixed dose according to the study protocol and it was not adjusted according to monitoring tests such as up the assessment for the factor Xa residual activity or thromboelastography [17,18].

Fifth, a large randomized controlled trial did not provide support of therapeutic anticoagulation in COVID-19 patients who are critically ill, since an initial strategy of therapeutic-dose anti-coagulation with heparin did not result in a greater probability of survival to hospital discharge or a greater number of days free of cardiovascular or respiratory organ support than did usual-care pharmacologic thromboprophylaxis [13]. Noteworthy, this trial included all COVID-19 patients admitted to ICU without defining any class of risk for thromboembolic complications; in addition, the median d-dimer value was around 2 times the upper normal limit [13], while in our study was > 4 as per inclusion criteria. This difference in the study population defines a population at increased risk for thromboembolic complications, that may have benefitted of therapeutic doses of anticoagulation on an individual basis [14]. Another multicenter randomized controlled trial, including patients with d-dimer levels >4 times of the upper normal limit, found that therapeutic-dose of enoxaparin reduced the primary composite outcome (i.e., major arterial and venous thromboembolism and all-cause mortality) compared with institutional standard heparin thromboprophylaxis among inpatients with COVID-19 [22]. However, the treatment effect was not seen in ICU patients [22]. In fact, the primary composite outcome occurred in 53.9% of ICU patients randomized to therapeutic dose of anticoagulants, and in 60.6% of patients in the control group (*p* = 0.560) [22]. Noteworthy, if we compute the same composite primary outcome in our population randomized to therapeutic enoxaparin, the rate of patients with major arterial and venous thromboembolism and all-cause mortality was 55.1%, very closed to findings by the previous study [22]. In addition, we surprisingly recorded that the 52% of patients in the enoxaparin group developed thromboembolism, that sounds to be a high proportion of patients. It should be noted that in our study ultrasonography was daily performed, whereas in the HEP-COVID trial only at day 10 or at hospital discharge and at a 30-days follow-up [22].

Lastly, it could be argued that we designed our study administering therapeutic dose of anticoagulation in critically ill COVID-19 patients, although considered at risk for thromboembolism and “against” the current guidelines [12,44,45]. It should be mentioned that we designed this study in 2020, as attested by the Ethical Committee approval date, but we could run the trial only 1 year later. Second, large clinical trials did not prove any superiority of therapeutic dose of heparin/enoxaparin over prophylactic doses [13,22]. In addition, even guidelines from the American Society of Hematology suggest using prophylactic over therapeutic-intensity anticoagulation for patients with COVID-19 related critical illness without suspected or confirmed venous thromboembolism [44]. However, the certainty in evidence is defined very low, because of serious imprecision, recalling the need for further studies [44]. The novelty was to test a different anticoagulant (i.e., bivalirudine) as compared to enoxaparin. Nevertheless, we could not find any clinical benefit of the former drug over the latter; we only further confirmed findings from previous studies in group of patients receiving therapeutic-intensity anticoagulation. In addition, we lack of a control group constituted by patients receiving prophylactic doses of anti-coagulation or no coagulants at all, although American Society of Hematology guidelines suggest using prophylactic anticoagulation in patients with COVID-19 related critical illness without suspected or confirmed venous thromboembolism [44]. Therefore, as suggested by current guidelines [12,44,45], therapeutic anticoagulation should not be considered as standard therapy in critically ill COVID-19 patients only at risk of thrombotic events, but only in case of confirmed thromboembolism.

## 5. Conclusions

In this small pilot study, bivalirudin did not reduce the time spent under invasive mechanical ventilation when compared to enoxaparin. In addition, the thromboembolic complications, the ICU mortality and length of stay and the 28-days survival were similar between treatment.

## Figures and Tables

**Figure 1 jcm-11-05992-f001:**
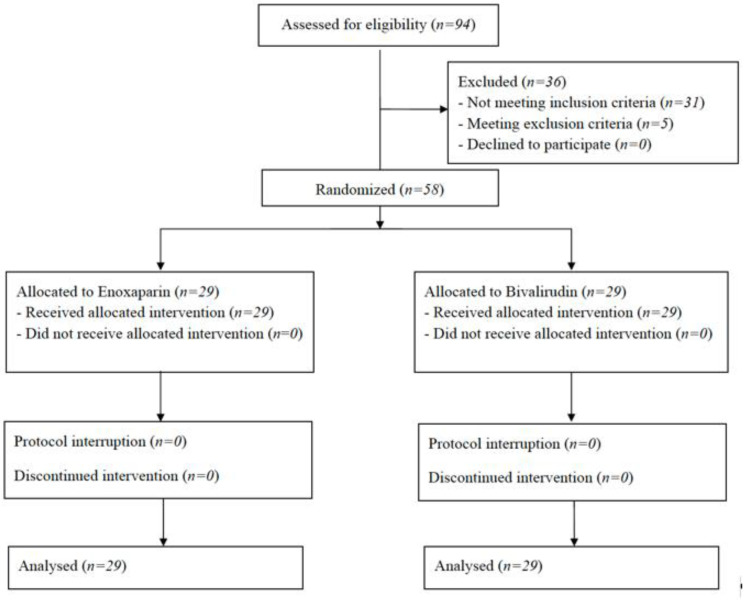
The figure depicts the study flowchart.

**Figure 2 jcm-11-05992-f002:**
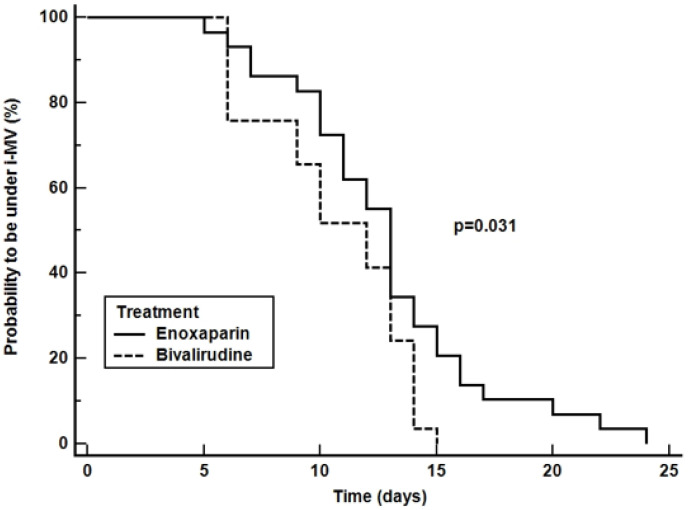
Probability to receive invasive mechanical ventilation during 28-day of follow-up. The curves depict the probability to receive invasive mechanical ventilation during 28-day of follow-up from randomization in patients receiving enoxaparin (solid line) or bivalirudin (dashed line).

**Figure 3 jcm-11-05992-f003:**
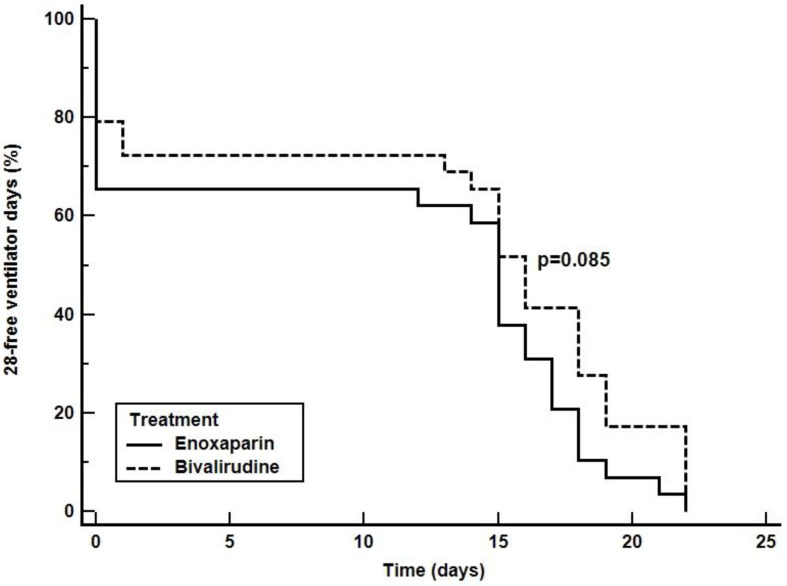
28-free ventilator days. The curves depict the probability to be weaned off mechanical ventilation during 28-day of follow-up from randomization in patients receiving enoxaparin (solid line) or bivalirudin (dashed line).

**Figure 4 jcm-11-05992-f004:**
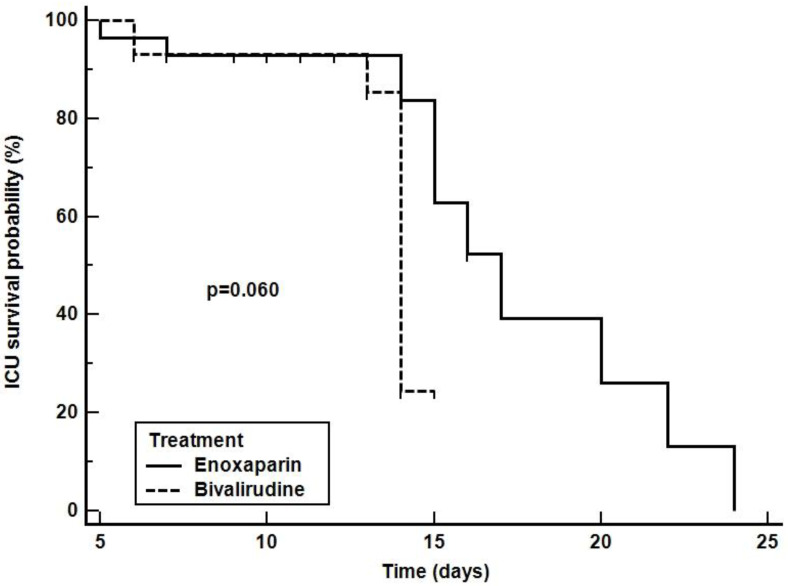
ICU survival. The curves depict the ICU survival in patients receiving enoxaparin (solid line) or bivalirudin (dashed line).

**Figure 5 jcm-11-05992-f005:**
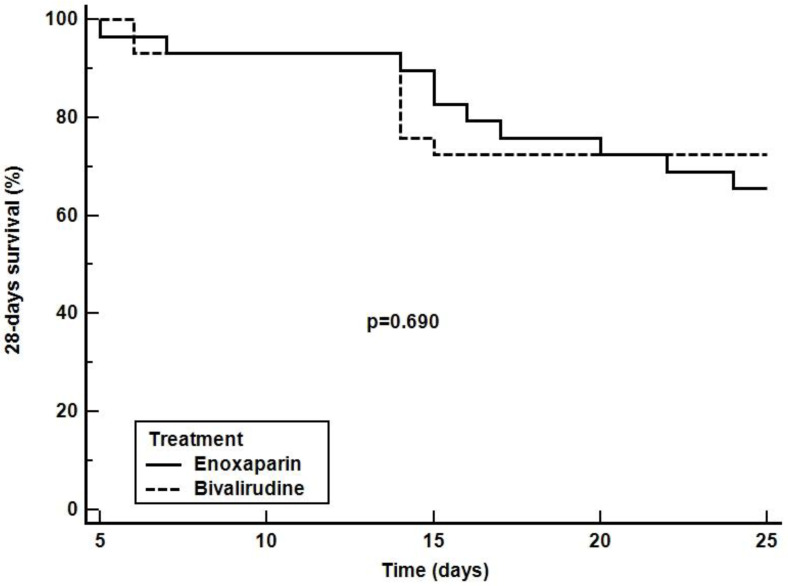
28-days survival rate. The curves depict the 28-day survival rate from randomization in patients receiving enoxaparin (solid line) or bivalirudin (dashed line).

**Table 1 jcm-11-05992-t001:** Characteristics of the study population.

	Bivalirudin (*n* = 29)	Enoxaparin (*n* = 29)	*p* Value
Age (years)	65 [58; 71]	60 [52; 69]	0.249
Male-*n* (%)	13 (44.8%)	18 (62.1%)	0.864
BMI (kg/m^2^)	27.7 [25.6; 31.6]	27.0 [24.7; 29.4]	0.591
SAPS-II	55 [50; 60]	56 [51; 60]	0.779
SOFA	11 [8; 12]	8 [7; 12]	0.117
Padua Score	6 [5; 6]	5 [5; 6]	0.076
Hemodynamic			
HR (beats/min)	70 [61; 89]	84 [60; 100]	0.256
MAP (mmHg)	89 [80; 96]	89 [83; 95]	0.669
Need for norepinephrine-*n* (%)	15 (51.7%)	11 (37.9%)	0.689
Norepinephrine (mcg*kg/min)	0.3 [0.2; 0.4]	0.4 [0.3; 0.5]	0.119
Mechanical ventilation			
Controlled Volume Mode	29 (100%)	29 (100%)	>0.999
PEEP (cmH_2_O)	10 [8; 10]	10 [8; 12]	0.254
Tidal Volume (ml)	440 [415; 475]	430 [405; 465]	0.988
Driving Pressure (cmH_2_O)	10 [9; 11]	10 [8; 11]	0.338
Static Compliance	48.9 [39.6; 56.1]	44.4 [39.6; 52.0]	0.371
Prone Position-*n* (%)	20 (69.0%)	19 (65.5%)	0.089
Arterial Blood Gases			
pH	7.37 [7.37; 7.42]	7.40 [7.37; 7.43]	0.287
PaCO2 (mmHg)	41.4 [38.3; 45.5]	39.2 [38.0; 45.3]	0.560
PaO2/FiO2	140 [124; 152]	143 [134; 165]	0.202
HCO3 (mMol/L)	24.3 [22.9; 26.1]	24.5 [23.9; 26.2]	0.455
Lac (mMol/L)	1.7 [1.0; 2.4]	1.5 [0.9; 2.4]	0.834
Blood tests			
Platelets count	248 [182; 314]	247 [163; 347]	0.726
aPTT	33 [30; 38]	33 [29; 36]	0.662
PT	12 [11; 13]	12 [11; 13]	0.949
INR	1.10 [1.02; 1.19]	1.10 [1.01; 1.17]	0.774
D-dimer	7.65 [4.25; 13.32]	8.03 [5.45; 14.24]	0.854
Fibrinogen	428 [357; 625]	548 [377; 743]	0.107
Procalcitonin	3.49 [0.18; 10.21]	1.72 [0.15; 7.04]	0.648
C-Reactive Protein	64.5 [25.0; 127.0]	78.2 [37.1; 138.5]	0.565
Comorbidities-*n* (%)			
Chronic Respiratory Failure	8 (27.6%)	5 (17.2%)	0.530
Cardiovascular disease	9 (31.0%)	6 (20.7%)	0.550
Arterial Hypertension	12 (41.4%)	11 (37.9%)	0.999
Diabetes	4 (13.8%)	3 (10.3%)	0.999
Hypothyroidism	2 (6.7%)	3 (10.3%)	0.999
Hyperthyroidism	3 (10.3%)	4 (13.8%)	0.999

**Table 2 jcm-11-05992-t002:** Secondary outcomes.

	Bivalirudin (*n* = 29)	Enoxaparin (*n* = 29)	*p* Value
Thrombo-embolic complications	5 (17.2)	15 (51.7)	0.012
Thrombotic complications	3 (10.3)	10 (34.5)	0.056
Embolic complications	2 (6.9)	5 (17.2)	0.423
Tracheostomy	4 (13.8)	6 (20.7)	0.730
Reintubation	4 (13.8)	4 (13.8)	0.999
28-free ventilation days (days)	15 [0; 17]	16 [1; 19]	0.162
ICU Length of Stay (days)	14 [12; 14]	15 [13; 15]	0.076
ICU Mortality *n* (%)	8 (27.6)	10 (34.5)	0.777
28-days mortality *n* (%)	8 (27.6)	10 (34.5)	0.777

## Data Availability

The authors will share all of the individual participant data collected during the trial after de-identification, to researchers who provide a methodologically sound proposal. The full protocol and raw data are available at longhini.federico@gmail.com.

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
