# Peer review of "Bivalirudin vs. Enoxaparin in Intubated COVID-19 Patients: A Pilot Multicenter Randomized Controlled Trial"

_jcm, 2022, doi:10.3390/jcm11205992_

Round 1

Reviewer 1 Report

The manuscript is well written. In my opinion, the topic of the study is interesting and relevant to the field. The conclusions are consistent with the results and address the main question. I recommend the manuscript  need to make appropriate corrections.

1.       The following sentences have typos that need to be corrected.

Nonetheless, in the literature it ha been reported that up to 50% patients has a high factor Xa

residual activity and may be resistant to enoxaparin [17-18], leading to thrombocytopenia,

arterial and venous thrombosis [19].”

„Daily, until ICU discharge, all patients underwent a color-coded Doppler ultrasonography

to assess the presence of thrombi in both superficial and deep veins of the upper

and lower limbs, and in the internal jugular veins.”

2.       Figures 4 and 5 have an unnecessary element.

3.       The Authors should unify the description of the group taking enoxaparin in the tables and figures, because 2 terms are used, i.e. LWMH and enoxaparin.

Author Response

The manuscript is well written. In my opinion, the topic of the study is interesting and relevant to the field. The conclusions are consistent with the results and address the main question. I recommend the manuscript need to make appropriate corrections.

We thank the Reviewer for his/her consideration of our manuscript.

1) The following sentences have typos that need to be corrected.

“Nonetheless, in the literature it ha been reported that up to 50% patients has a high factor Xa residual activity and may be resistant to enoxaparin [17-18], leading to thrombocytopenia, arterial and venous thrombosis [19].”

Done.

“Daily, until ICU discharge, all patients underwent a color-coded Doppler ultrasonography to assess the presence of thrombi in both superficial and deep veins of the upper and lower limbs, and in the internal jugular veins.”

We apologize with the Reviewer, but we cannot find any typo errors. Please indicate it.

2) Figures 4 and 5 have an unnecessary element.

We thank with the Reviewer for his/her point and we have modified the Figures accordingly.

3) The Authors should unify the description of the group taking enoxaparin in the tables and figures, because 2 terms are used, i.e. LWMH and enoxaparin.

Done, thanks.

Reviewer 2 Report

This article reports a comparison study about the efficacy of anticoagulation with Bivalirudin or with Low Molecular Weight Heparin (LMWH) / Enoxaparin in a group of severe Covid-19 patients intubated. The survey lasts for 28 days and end-points cocern occurrence of thrombotic or embolic events, duration of intubation, stay in the ICU or mortality. Comparatively toformer studies evaluating the possible benefits of bivalirudin to prevent thrombo-embolic complications in Covid-19 patients, this study is performed on 2 randomized group of patients, with similar patterns of morbidity, disease severity and complications outcomes, although the total number in each group is low for the muktiple parameters analyzed, which weakens the statistical significance.  The authors did not find any statistical difference between both anticoagulation protocols for the end-points selected, but as they analyze it in the discussion section a more extensive study with a higher group of patients would be necessary to highlight portentiial differences. Therefore, this study illustrates only the authors experience with both anticoagulants, Bivalirudin and Enoxaparin, and shows that although anticoagulation remains highly beneficial in severe Covid-19 patients to limit the clinical consequences of the prothrombotic and hyper-inflammatory state, the lilmited data generated do not allow to demonstrtae any statistical superiory for Bivaluridin as compared with Enoxaparin. However, the incidence of thrombotic or embolic events looks lower in the Bivalirudin group but the limited number of patients in each group does not allow extracting any statistical significance. This should however deserve to be further investigated by including much more patients. Maybe, grouping the thrombotic and embolic groups in only one, there are then 5 events in the Bivalirudin group versus 15 in the Enoxaparin one, and this could become statiscally signifocant. I suggest to the authors to explore this possibility as the raw figures strongly suggest a significant difference in the thrombo-embolic events rate.

Concerning the specific comments, they are outlined here below:In abstract, the sentence just before "Results" requires attention at " ...ICU and 28-days mortaility"

In "background" after reference [16], it must be "has" and not "ha".

What does the authors mean by "... high Factor Xa residual activity and maybe resistant to enoxaparin"? Do they suggest that immuno-thrombosis and platelet activation, with a release of high amounts of PF4 can neutralize (at least partly) enoxaparin? Or do they mean that heparin resistance results from lack of AT?

In materials and methods, the reagents used, especially for PT, APTT and DDimer must be indicated, as clotting times or variations can be reagent brand dependent.

Legend to figure 2 must be revised, as it is confusing. It shows ventilation rate during the 28-days of follow-up, not after.The same in the text, just below this legend.

Legend to figure 4 must also be revise and completed. What do the authors mean by "ICU survival probabilit"?

Author Response

This article reports a comparison study about the efficacy of anticoagulation with Bivalirudin or with Low Molecular Weight Heparin (LMWH) / Enoxaparin in a group of severe Covid-19 patients intubated. The survey lasts for 28 days and end-points concern occurrence of thrombotic or embolic events, duration of intubation, stay in the ICU or mortality. Comparatively to former studies evaluating the possible benefits of bivalirudin to prevent thrombo-embolic complications in Covid-19 patients, this study is performed on 2 randomized group of patients, with similar patterns of morbidity, disease severity and complications outcomes, although the total number in each group is low for the multiple parameters analyzed, which weakens the statistical significance.  The authors did not find any statistical difference between both anticoagulation protocols for the end-points selected, but as they analyze it in the discussion section a more extensive study with a higher group of patients would be necessary to highlight potential differences. Therefore, this study illustrates only the authors experience with both anticoagulants, Bivalirudin and Enoxaparin, and shows that although anticoagulation remains highly beneficial in severe Covid-19 patients to limit the clinical consequences of the prothrombotic and hyper-inflammatory state, the limited data generated do not allow to demonstrate any statistical superiority for Bivaluridin as compared with Enoxaparin.

We thank the Reviewer for his/her positive comment on our study.

1) However, the incidence of thrombotic or embolic events looks lower in the Bivalirudin group but the limited number of patients in each group does not allow extracting any statistical significance. This should however deserve to be further investigated by including much more patients. Maybe, grouping the thrombotic and embolic groups in only one, there are then 5 events in the Bivalirudin group versus 15 in the Enoxaparin one, and this could become statiscally signifocant. I suggest to the authors to explore this possibility as the raw figures strongly suggest a significant difference in the thrombo-embolic events rate.

We thank the Reviewer for his/her suggestion We have followed this indication and reported a significant difference between groups in thrombo-embolic events when analyzed together.

2) Concerning the specific comments, they are outlined here below:

In abstract, the sentence just before "Results" requires attention at " ...ICU and 28-days mortaility"

Done, thanks.

3) In "background" after reference [16], it must be "has" and not "ha".

Done, thanks.

4) What does the authors mean by "... high Factor Xa residual activity and maybe resistant to enoxaparin"? Do they suggest that immuno-thrombosis and platelet activation, with a release of high amounts of PF4 can neutralize (at least partly) enoxaparin? Or do they mean that heparin resistance results from lack of AT?

Enoxaparin has an anti-Xa factor activity. As reported by the quoted literature, the prothrombotic status may be related to an increased blood concentration of Factor Xa. Therefore, the action of enoxaparin may be reduced and may induce some alteration of the coagulation. Therefore, this is out of conditions such as platelet activation mediated by PF4 or lack of AT-3, that prevents the action by sodium heparin.

5) In materials and methods, the reagents used, especially for PT, APTT and DDimer must be indicated, as clotting times or variations can be reagent brand dependent.

Following the Reviewer’s indication, we now specify the system for testing coagulation.

6) Legend to figure 2 must be revised, as it is confusing. It shows ventilation rate during the 28-days of follow-up, not after. The same in the text, just below this legend.

Done, thanks.

7) Legend to figure 4 must also be revise and completed. What do the authors mean by "ICU survival probabilit"?

The legend has been modified. Thanks.